# MaPom1, a Dual-Specificity Tyrosine Phosphorylation-Regulated Kinase, Positively Regulates Thermal and UV-B Tolerance in *Metarhizium acridum*

**DOI:** 10.3390/ijms252211860

**Published:** 2024-11-05

**Authors:** Yu Zhang, Lei Song, Yuxian Xia

**Affiliations:** 1School of Life Sciences, Chongqing University, Chongqing 401331, China; 18838933368@163.com (Y.Z.); songlei19960404@163.com (L.S.); 2Chongqing Engineering Research Center for Fungal Insecticides, Chongqing 401331, China; 3Key Laboratory of Gene Function and Regulation Technologies Under Chongqing Municipal Education Commission, Chongqing 401331, China; 4National Engineering Research Center of Microbial Pesticides, Chongqing 401331, China

**Keywords:** *Metarhizium acridum*, MaPom1 protein kinase, thermotolerance, heat shock proteins, UV-B tolerance, DNA damage and repair

## Abstract

Fungi play irreplaceable roles in the functioning of natural ecosystems, but global warming poses a significant threat to them. However, the mechanisms underlying fungal tolerance to thermal and UV-B stresses remain largely unknown. Dual-specificity tyrosine phosphorylation-regulated kinase (DYRK) Pom1 is crucial for fungal growth, conidiation, and virulence. However, its role in stress tolerance within kingdom fungi has not been explored. In this study, we analyzed the function of *MaPom1* (a Pom1 homologous gene) in the entomopathogenic fungus *Metarhizium acridum* and its regulatory roles in stress tolerance. Conidial thermal and UV-B tolerance significantly decreased in the *MaPom1* disruption strain (Δ*MaPom1*), whereas conidial yield and virulence were unaffected. RNA-Seq analysis indicated that the differentially expressed genes (DEGs) were primarily related to amino sugar, nucleotide sugar metabolism, cell wall components, growth and development, and stress response pathways. Under heat shock treatment, the expression levels of heat shock protein genes decreased significantly, leading to reduced thermotolerance. Moreover, under UV-B treatment, *MaPom1* expression and the enzyme activity significantly changed, indicating its involvement in regulating UV-B tolerance. The percentage of nuclear damage in Δ*MaPom1* under UV-B treatment was higher than that in the wild-type strain (WT) and the complementary strain (CP). Additionally, the transcription levels of DNA damage-related genes significantly decreased, whereas those of several genes involved in the DNA damage repair response increased significantly. Overall, *MaPom1* contributed to thermal and UV-B tolerance by regulating the expression of heat shock protein genes and DNA damage repair genes.

## 1. Introduction

Global warming and climate change have resulted in significant abiotic stresses on life and significantly affect the global biogeochemical cycle [1,2]. By 2050, rising temperatures may cause changes in fungal and insect populations and reduce crop yields [3,4]. Fungi often show greater tolerance to stress, and fungal diversity in extreme environments is higher than previously thought [5]. Entomopathogenic fungi, as environment-friendly biopesticides, play crucial roles in agricultural pest control [6] and regulate insect populations [7] with low potential to induce insect resistance [8,9]. However, temperature and UV-B variations, as adverse environmental stressors, profoundly influence the efficacy of entomopathogenic fungi in pest management strategies [10,11,12,13,14] and insect population control [15]. These restrictions may render biological control strategies ineffective and limit their application in agriculture [16,17,18]. Therefore, it is essential to investigate the molecular mechanisms underlying tolerance to heat shock and UV-B for promoting the effectiveness and durability of fungal biopesticides in pest management.

Conidia are responsible for fungal dispersal and environmental persistence. In pathogenic fungi, conidial production, survival, dispersal, and pathogenicity are strongly affected by exposure to solar radiation. UV radiation can also reduce conidial germination speed and virulence [19]. Temperature directly affects fungal growth and protein dynamics, ultimately impacting overall fungal performance [20]. DNA serves as the primary chromophore for absorbing sunlight energy. The exposure of genomic DNA to ultraviolet radiation can cause DNA damage [21]. Adverse environmental temperatures further affect fungal survival by inducing protein misfolding and DNA damage [22,23]. Over the past 50 years, the mechanisms underlying fungal DNA damage repair have been extensively studied [24]. In *Aspergillus nidulans*, DNA repair can be triggered by responses to harmful environmental changes [25]. Heat shock proteins (HSPs) are ubiquitous and conserved protein families in organisms that play crucial roles in maintaining cellular proteostasis, protecting cells from stress [26], and facilitating processes such as transcriptional, translational, and posttranslational modification, protein conformational changes, and the prevention of protein aggregation and depolymerization [27]. Under heat shock conditions, *Beauveria bassiana* coordinates gene expression and chromatin structure via heat shock factor 1 (HSF1) and Hsp90, resulting in heat adaptation and changes in virulence [28]. In *Aspergillus*, HSF1 regulates the heat shock response and expression of HSPs while also enhancing heat tolerance by regulating cell wall biosynthesis, remodeling, and the expression of genes related to lipid balance [29]. Similar to *Aspergillus*, the pigment properties of the spores in *Metarhizium anisopliae* are related to reduced UV-B and temperature tolerance [30]. Overall, the mechanisms by which entomopathogenic fungi develop tolerance to temperature and UV-B radiation have not been extensively studied, and knowledge at the phenotypic and genetic levels remains limited.

Protein phosphorylation is a vital posttranslational modification process that involves the substrate subcellular of protein distribution [31] and the incorporation of phosphate groups into specific amino acids within protein polypeptide sequences [32], which play important roles in intracellular signal transduction [33]. Serine/threonine protein kinases use ATP as a phosphate donor to catalyze the phosphorylation of serine or threonine residues in target proteins. These kinases play crucial roles in various signal transduction pathways and cellular processes, including cellular triggering [34], amplification, apoptosis [35], programmed cell death (PCD) [36], and self-digestion [37,38,39]. The DYRK family proteins are phylogenetically conserved [40]. In humans, DYRK1A affects cancer by regulating various biological processes [40,41,42].

In fungi, the serine/threonine protein kinase Pom1 of the DYRK family has been extensively studied for cell growth and division, polar growth, and cellular localization [43]. Two MaPom1 homologous proteins have been widely explored in yeast. Yak1p in *Saccharomyces cerevisiae* is associated with the regulation of the cell cycle, cytokinesis, and cell differentiation [44], whereas Pom1p in *Schizosaccharomyces pombe* is involved in cell division and growth [45,46]. Pom1 acts as a direct sensor to control cell size at the onset of mitosis [47]. In rod-shaped fission yeast, Pom1 localizes at the cell tip, and entry into mitosis is delayed until the cell reaches a critical size [43,48,49]. Similarly, in *Dictyostelium discoideum*, the protein kinase YakA controls the cell cycle and regulates cell division intervals [50]. In *Aspergillus niger*, PomA kinase functions as an inhibitory factor for SepH (Cdc7p) kinase and negatively regulates conidial separation and formation [51]. In the phytopathogenic fungus *Colletotrichum scovillei*, Pom1 negatively regulates cell division. Asexual growth, surface hydrophobicity, and conidial capacity decreased under osmotic stress in Δ*CsPom1*. Additionally, the Δ*CsPom1* mutant exhibited delayed penetration and invasive growth, leading to a significant reduction in virulence [52]. Overall, Pom1 plays crucial roles in the cell growth [53], mitosis [54], osmotic stress tolerance [52], and virulence [55] of various fungi. However, the functions of Pom1 and its regulatory mechanisms in thermotolerance and UV-B tolerance have not been investigated in *Metarhizium acridum*.

In this study, we examined the functions and related mechanisms of *MaPom1* in *M. acridum*. The findings indicated that Δ*MaPom1* germinated earlier and its conidiation was delayed, with a 50% reduction in mycelial branches and septa. However, conidial yield and virulence were not affected. Additionally, the thermal and UV-B tolerance of Δ*MaPom1* conidia decreased significantly. Reduced thermotolerance may be related to the expression levels of heat shock protein genes, whereas decreased UV-B tolerance was induced by impaired DNA damage repair mechanisms.

## 2. Results

### 2.1. Characteristics of MaPom1

The total length of *MaPom1* is 4122 bp, including a 60 bp intron. MaPom1 kinase consists of 1353 amino acids and has an isoelectric point of 10.08. The MaPom1 protein is composed of three domains: PHA03247, PHA03307, and the catalytic domain of the bispecific protein kinase Pkc_DYRK (Figure 1A). The alignment of the conserved catalytic domain (Pkc_DYRK) revealed that MaPom1 was homologous to kinases found in various fungi (Figure 1B). Phylogenetic tree analysis showed that MaPom1 in *M. acridum* clustered with Pom1 in *M. robertsii* (Figure 1C). To explore the function of MaPom1 in *M. acridum*, we constructed a pK2-PB-*MaPom1*-LR knockout vector and its corresponding complemented vector, pK2-Sur-*MaPom1*-eGFP (Figure 2A). Southern blotting was used to confirm the successful generation of the knockout and complemented strains (Figure 2B). To determine whether MaPom1 acts as a serine/threonine protein kinase in *M. acridum*, we measured the serine/threonine protein kinase activity. The results revealed that the enzyme activity of Δ*MaPom1* was lower than that of the WT and CP (F_2,6_ = 243.3, *p* < 0.0001; WT vs. Δ*MaPom1*, *p* < 0.0001; WT vs. CP, *p* = 0.0026; Δ*MaPom1* vs. CP, *p* < 0.0001; Tukey’s test, Figure 2C), confirming that MaPom1 serves as a serine/threonine protein kinase.

### 2.2. MaPom1 Contributes to Conidial Germination and Mycelial Growth with No Effect on Conidial Yield and Virulence

To investigate the impact of *MaPom1* on the growth and germination of *M. acridum*, we observed septa and branches after 18 h of cultivation (Figure 3A). The results indicated that the number of septa in Δ*MaPom1* (6.67 ± 0.58) was significantly lower than that in WT (14.00 ± 1.00) and CP (13.67 ± 1.15) (F_2,6_ = 57.88, *p* = 0.0001; WT vs. Δ*MaPom1*, *p* = 0.0002; WT vs. CP, *p* = 0. 9034; Δ*MaPom1* vs. CP, *p* = 0.0002; Tukey’s test, Appendix A). Similarly, the number of branches in Δ*MaPom1* (4.00 ± 1.00) was significantly lower than that in WT (8.00 ± 1.00) and CP (7.67 ± 0.58) (F_2,6_ = 19.00, *p* = 0.0025; WT vs. Δ*MaPom1*, *p* = 0.0035; WT vs. CP, *p* = 0. 8906; Δ*MaPom1* vs. CP, *p* = 0.0054; Tukey’s test, Appendix A), indicating that *MaPom1* affected mycelial division. We also evaluated germination rates at 2, 4, 6, 8, 10, and 12 h and found that Δ*MaPom1* had a higher germination rate than WT and CP on 1/4 SDAY medium (Figure 3B). The half-germination time (GT_50_) of Δ*MaPom1* (3.75 ± 0.03 h) was significantly shorter than that of WT (6.13 ± 0.62 h) and CP (5.56 ± 0.10 h) (F_2,6_ = 35.36, *p* = 0.0005; WT vs. Δ*MaPom1*, *p* = 0.0005; WT vs. CP, *p* = 0.2068; Δ*MaPom1* vs. CP, *p* = 0.0021; Tukey’s test, Figure 3C), suggesting that *MaPom1* regulates conidial germination and mycelial growth. Furthermore, the conidia yield of Δ*MaPom1* significantly decreased by 30% compared to WT and CP on the third day, whereas no significant differences in conidial yield were detected after 3 d (Appendix A). Δ*MaPom1* also exhibited a longer mycelium (Figure 3D) that delayed conidiation (Appendix A). Virulence assays revealed that the survival curves of locusts infected with WT, Δ*MaPom1*, and CP exhibited similar trends (Figure 3E), and the half-lethal time (LT_50_) showed no significant difference in the topical assay (F_2,6_ = 3.160, *p* = 0.1150; WT vs. Δ*MaPom1*, *p* = 0.1150; WT vs. CP, *p* = 0.8390; Δ*MaPom1* vs. CP, *p* = 0.2377; Tukey’s test, Figure 3F), suggesting that *MaPom1* did not influence the virulence of *M. acridum*.

### 2.3. MaPom1 Is Essential for Thermal and UV-B Tolerance

To determine whether *MaPom1* affects conidial stress tolerance, we measured the conidial germination rate under heat shock (45 °C) and UV-B treatments. The results indicated a significant decrease in both thermal and UV-B tolerance (Figure 4A,D). The half-inhibition time (IT_50_) for Δ*MaPom1* (3.89 ± 0.25 h) was significantly shorter than that of WT (6.21 ± 0.24 h) and CP (6.31 ± 0.43 h) under heat shock treatment (F_2,6_ = 63.40, *p* < 0.0001; WT vs. Δ*MaPom1*, *p* = 0.0002; WT vs. CP, *p* = 0.9071; Δ*MaPom1* vs. CP, *p* = 0.0001; Tukey’s test, Appendix A). There was no difference in the relative expression levels of *MaPom1* under heat shock treatment in WT (F_2,6_ = 0.4471, *p* = 0.6592; Heat-shock-0 h vs. Heat-shock-3 h, *p* = 0.8223; Heat-shock-0 h vs. Heat-shock-6 h, *p* = 0.9437; Heat-shock-3 h vs. Heat-shock-6 h, *p* = 0.6420; Tukey’s test, Figure 4B). Serine/threonine protein kinase activity was significantly decreased in Δ*MaPom1*, although it remained unchanged after 3 and 6 h of heat shock treatment (Figure 4C), suggesting that *MaPom1* may regulate thermal tolerance by affecting other factors. Δ*MaPom1* also exhibited lower UV-B tolerance compared to WT and CP (Figure 4D), with IT_50_ values of 6.40 ± 0.11 h, 4.91 ± 0.20 h, and 6.73 ± 0.10 h, respectively (F_2,6_ = 74.38, *p* < 0.0001; WT vs. Δ*MaPom1*, *p* = 0.0002; WT vs. CP, *p* = 0.1713; Δ*MaPom1* vs. CP, *p* < 0.0001; Tukey’s test, Appendix A). *MaPom1* expression was significantly upregulated in WT after UV-B irradiation (F_2,6_ = 412.4, *p* < 0.0001; UV-B-0 h vs. UV-B-3 h, *p* < 0.0001; UV-B-0 h vs. UV-B-4.5 h, *p* < 0.0001; UV-B-3 h vs. UV-B-4.5 h, *p* = 0.9125; Tukey’s test, Figure 4E). In contrast, serine/threonine protein kinase activity in Δ*MaPom1* was notably decreased compared to WT and remained consistently lower under UV-B treatment (Figure 4F). These results indicated *MaPom1* played a curial role in regulating UV-B tolerance at the transcriptional level.

### 2.4. Identification of DEGs Influenced by MaPom1

To further investigate the mechanism by which *MaPom1* regulates stress tolerance, RNA-seq analysis was performed to compare the DEGs in 3-day-old conidia of Δ*MaPom1* and WT (the earliest time point at which stress tolerance phenotypes appeared) on 1/4 SDAY medium (Appendix A). The RNA-seq results identified 85 DEGs in Δ*MaPom1* compared to WT (|log_2_ ratio| ≥ 2, q ≤ 0.05) (Appendix A), with 41 upregulated and 44 downregulated (Figure 5A). To validate the digital expression profiling, 17 DEGs were selected for expression level determination using qRT-PCR (Appendix A). The qRT-PCR results confirmed that 16 of the 17 DEGs exhibited expression patterns consistent with those obtained from RNA-seq, except for MAC_09740, indicating the reliability of the RNA-seq results (Appendix A). Sugar metabolism plays a curial role in regulating the synthesis of cell wall polymers [56]. KEGG pathway analysis revealed the significant enrichment of DEGs in processes related to primary metabolisms, such as amino sugar metabolism, nucleotide sugar metabolism, steroid biosynthesis, peroxisomal functions, α-linolenic acid metabolism, and the biosynthesis pathways of pantothenate and CoA (Figure 5C), all of which may affect stress tolerance. The DEGs were also enriched in 16 GO terms, including six related to biological processes, four related to cellular components, and six related to molecular functions (Figure 5C). The cell wall is a curial component of fungal homeostasis [57]. These GO terms are associated with cell wall and membrane components as well as cell structure and function, which may contribute to reduced stress tolerance.

Further KEGG and GO pathway analyses revealed that the DEGs were primarily related to growth and development (MAC_06007, MAC_09143, and MAC_06365) and other biological activities (Appendix A), which could lead to slower germination and fewer branches. Late embryogenesis abundant (LEA) domain-containing proteins provide organisms with evolutionary advantages under various stress environments [58]. Notably, the LEA domain-containing protein gene (MAC_00250) was significantly downregulated, potentially contributing to the reduced stress tolerance. The fungal cell wall, which serves as a physical barrier, is closely related to cell survival, pathogenicity, and resistance [59]. Genes related to cell wall components (MAC_03963, MAC_09810, and MAC_05133) were significantly upregulated in Δ*MaPom1*, suggesting that the reduced stress tolerance may not be directly related to alterations in the cell wall composition.

### 2.5. MaPom1 Contributes to Stress Tolerance Independently of Cell Wall Composition

KEGG and GO annotation analyses revealed that the DEGs were primarily related to cell wall components, including upregulated genes encoding membrane transporter proteins (MAC_09810) and cell wall proteins (MAC_05133), as well as a downregulated gene encoding hydrophobin (MAC_09507). To determine whether the reduced thermal and UV-B tolerance observed in Δ*MaPom1* was linked to cell wall composition and integrity, we assessed the sensitivity of fungal strains to cell wall-disrupting agents. The results showed no significant changes in sensitivity (Figure 6A), indicating that stress tolerance reduction was not related to the cell wall. Related studies have suggested that the increased stress tolerance in conidia is associated with trehalose accumulation [60]. Although RNA-Seq data showed that putative trehalose-6-phosphate synthase/trehalose phosphatase (MAC_09143) was upregulated (Appendix A), the trehalose content in Δ*MaPom1* was not significantly different from that in the WT and CP (F_2,6_ = 4.839, *p* = 0.0561; WT vs. Δ*MaPom1*, *p* = 0.1234; WT vs. CP, *p* = 0.8315; Δ*MaPom1* vs. CP, *p* = 0.0586; Tukey’s test, Figure 6B). These results indicated that the decreased thermal and UV-B tolerance in Δ*MaPom1* was not related to trehalose metabolism.

### 2.6. MaPom1 Contributes to UV-B Tolerance by Regulating DNA Damage Repair

RNA-seq analysis indicated that that reduced UV-B tolerance in Δ*MaPom1* may be linked to impaired DNA damage repair. To explore this further, we assessed DNA damage after UV-B and heat shock treatments. Δ*MaPom1* exhibited significant nuclear dispersion after 3 and 4.5 h of UV-B exposure (Figure 7A), with nuclear dispersion rates reaching 60% and 80%, respectively, which were significantly higher than these in the WT and CP (Figure 7B). Furthermore, WT and CP demonstrated DNA repair after 3 h of recovery cultivation, whereas Δ*MaPom1* showed no significant DNA repair (Figure 7C,D), indicating that the reduced UV-B tolerance of Δ*MaPom1* is associated with defects in DNA damage repair. To confirm the involvement of DNA damage repair mechanisms, we measured the expression of DNA damage repair-related genes after UV-B exposure by qRT-PCR. Damage-specific DNA binding-protein 1 (DDB1) is a multifunctional protein that recognizes UV-induced DNA damage and plays a crucial role in regulating UV-induced gene transcription. Additionally, DDB1 acts as a damage sensor, helping to maintain genomic integrity and ensuring proper cell cycle progression [61]. The expression levels of a putative DNA-binding protein (MAC_06589), putative O-methyltransferase (MAC_09811), and putative NAD(P)H-dependent oxidoreductase (MAC_05938) were significantly decreased in Δ*MaPom1* (Figure 7E), which contribute to its reduced UV-B tolerance. Additionally, high mobility group (HMG) box proteins [62], RNA-binding proteins [63,64], DNA PKcs [65], and flavin adenine dinucleotide (FAD) cofactors [66] are known to participate in the DNA damage response (DDR). In Δ*MaPom1*, the expression levels of FAD-dependent oxidoreductase family protein (MAC_00180), protein kinase with a PKc-like catalytic domain (MAC_05043), putative RNA-binding protein (MAC_05343), and putative HMG box protein (MAC_04576) were significantly increased (Figure 7E), further supporting the role of *MaPom1* in regulating DNA damage repair-related genes in response to UV-B stress. However, DAPI staining after heat shock treatment revealed no significant differences in DNA damage (Appendix A), suggesting that decreased thermotolerance was not related to DNA damage. Heat shock proteins (HSPs), which are highly conserved and expressed under stress conditions to protect cells [24], were significantly downregulated in Δ*MaPom1* after 3 h and 6 h of heat shock treatment (Appendix A), indicating that reduced thermotolerance may be linked to decreased HSP expression.

## 3. Discussion

### 3.1. MaPom1 Affects Conidial Germination but Not Virulence and Conidial Yield

To date, numerous studies have explored the functions of Pom1 in fungi. Pom1 regulates cell division and polar growth in yeast [46,67]. Specifically, the DYRK kinase Pom1 is involved in cell division and suppresses cellular proliferation at the cellular terminus in fission yeast [54,68]. Cytoplasmic division is essential for hyphal growth in filamentous fungi [69]. In this study, the reduced number of branches and septa in Δ*MaPom1* was consistent with the phenotype observed in yeast [67]. In filamentous fungi, conidia play crucial roles in asexual reproduction and environmental dissemination [70,71,72]. Here, the absence of *MaPom1* caused early germination and delayed conidiation without affecting conidial yield or virulence in *M. acridum*. In constrast, the absence of *CsPom1* in *C. scovillei* significantly reduced conidial production, virulence, septum formation, conidial size and shape, and cell division [52], suggesting that the effects of Pom1 on virulence and conidial yield are not conserved across filamentous fungi.

### 3.2. MaPom1 Positively Regulates UV-B and Thermal Tolerance Not via Cell Wall Composition

Natural stress factors can affect the survival of entomopathogenic fungi in the environment [73]. Ultraviolet radiation can reduce fungal spore viability, and heat can lower conidial germination rates [13,74]. In this study, the conidial germination rates of Δ*MaPom1* decreased significantly under heat shock and UV-B treatments compared to WT. The fungal cell wall maintains cellular homeostasis [75] and responds to environmental pressure, thereby affecting fungal stress tolerance [76]. Here, the sensitivity of Δ*MaPom1* to cell wall-disrupting agents has not changed compared to WT and CP. Trehalose, a key stress metabolite, protects fungi from extreme environmental conditions [60,77,78,79] and plays an essential role in promoting productivity, enhancing stress biology, and influencing virulence [80]. However, the trehalose content in Δ*MaPom1* was not significantly different from that in WT and CP, indicating that *MaPom1* regulates stress tolerance not via cell wall composition. Numerous genes have been reported to influence stress tolerance in *M. acridum* [81,82]. For instance, *MaSep1* negatively affects fungal tolerance to heat shock and UV-B irradiation by regulating the expression of genes related to DNA repair and ROS clearance [83]. In contrast, *MaAts* enhances fungal tolerance to UV-B irradiation and heat shock by affecting the transcription of genes involved in melanin synthesis, cell wall integrity, and various stress tolerance mechanisms [84]. These findings suggest that different genes regulate stress tolerance in *M. acridum* through distinct pathways.

### 3.3. MaPom1 Affects UV-B Tolerance via Regulating the Transcription Level of DNA Damage and Repair Genes

Environmental stressors such as heat shock and UV-B radiation, can cause DNA damage and impede DNA repair [85]. DNA is a primary target of UV-B-induced damage, and cells have developed various repair or tolerance mechanisms to counteract this damage, including light repair, excision repair, mutation repair, recombination repair, cell cycle checkpoints, and apoptosis [86]. The DNA-dependent protein kinase catalytic subunit (DNA-PKcs) is a vital component of the DNA-PK complex, which is crucial for the repair of DNA double-strand breaks via non-homologous end-joining pathways [87]. In this study, UV-B irradiation caused DNA damage in Δ*MaPom1* and reduced its ability to recover from DNA damage. In addition, RNA-binding proteins (RBPs) have been shown to significantly influence the DNA damage response (DDR) and DNA repair [88,89], either by participating in DDR [69,70] or by regulating the expression of DDR-related genes [90]. High mobility group box protein 1 (HMGB1) serves as an architectural protein and plays a pivotal role in the assembly of protein–DNA complexes implicated in transcriptional processes, genetic recombination, DNA repair mechanisms, and chromatin structural modifications [91]. The expression levels of DNA damage and repair-related genes were significantly affected in Δ*MaPom1* (Appendix A), suggesting that the reduced UV-B tolerance of Δ*MaPom1* was due to the modulation of genes related to DNA damage and repair, a mechanism similar to the regulatory mechanism in Δ*MaSep1* [83]. To sum up, UV-B radiation could cause DNA damage, and the transcription level of DNA damage genes were upregulated, and DNA repair genes downregulated, in Δ*MaPom1*, indicating that *MaPom1* regulated UV-B tolerance via the DNA damage and repair pathway.

### 3.4. MaPom1 Affects Thermal Tolerance via Regulating the Transcription Level of HSPs

Active fungi respond to stress by producing HSPs and chaperones that facilitate the repair of damaged functional structures [27,92]. Stress often leads to protein misfolding at the cellular level, and HSPs are activated to minimize the accumulation of denatured or abnormal proteins [93]. HSPs respond to environmental stresses by binding to heat shock elements (HSEs), promoting HSP translation, and preventing cellular damage [94]. Heat shock triggers complex cellular responses, including HSP induction, RNA and DNA damage, abnormal protein degradation, and disruptions in cell cycle progression, which ultimately cause cell death [95]. HSPs are highly conserved across organisms and expressed under stressful conditions to protect cells from damage [26,96]. They can serve as biomarkers for assessing cell damage in high-temperature environments [97]. By refolding or degrading misfolded proteins, HSPs play a critical role in repairing stress-induced protein damage [98]. For instance, in *Aspergillus fumigatus*, Hsp90 is transcriptionally regulated in response to heat shock [99]. In this study, the expression levels of HSPs in Δ*MaPom1* were significantly decreased after heat shock treatment, suggesting that HSP gene expression contributed to the diminished heat shock tolerance of the strain. In summary, *MaPom1* regulates thermal tolerance via reducing the transcription level of HSPs, which makes a small amount of HSP to protect cells from damage in high-temperature environments.

## 4. Materials and Methods

### 4.1. Strains and Cultivation Conditions

The wild-type strain (WT), designated CQMa102, was obtained from the China General Microbiological Culture Collection Center (CGMCC; No. 0877). All fungal strains used in this study were cultivated on a medium comprising one-quarter-strength Sabouraud’s dextrose agar medium (1/4 SDAY). Plasmid construction was performed using *Escherichia coli* BGT1 obtained from BioGround Biotechnology (Chongqing, China). Fungal transformation was performed by utilizing *Agrobacterium tumefaciens* AGL-1 [100], which was sourced from Weidi Biotechnology in Shanghai, China.

### 4.2. Bioinformatic Analysis

All homologous genes and protein sequences of *MaPom1* (GenBank accession numbers: MAC_02617 (gene) and XP_007808957.1 (protein)) were downloaded from the NCBI website (https://www.ncbi.nlm.nih.gov/ (accessed on 14 February 2022)). The *MaPom1* protein domain was identified using the SMART interface (http://smart.embl.de/ (accessed on 14 February 2022)). The DNAMAN software (https://www.lynnon.com/dnaman.html (accessed on 14 February 2022), Lynnon Biosoft, San Ramon, CA, USA) was used for multiple sequence alignment. A neighbor-joining phylogenetic tree was generated using MEGA 7.0 software (http://www.megasoftware.net/ (accessed on 14 February 2022)), supported by 1000 bootstrap replicates for statistical robustness [101].

### 4.3. Construction of MaPom1 Mutants

The left and right arms of the *MaPom1* gene were amplified using the primers *MaPom1*-LF/*MaPom1*-LR and *MaPom1*-RF/*MaPom1*-RR (Appendix A), respectively, and inserted into the pK2-PB vector, which contains a phosphinothricin-resistant bar gene, to facilitate the generation of a deletion vector. The obtained vector was transformed into *M. acridum* via *A. tumefaciens* [102]. Positive transformants were screened on Czapek–Dox medium supplemented with 500 μg/mL glufosinate ammonium (Sigma-Aldrich, St. Louis, MO, USA). The *MaPom1* sequence with its promoter (6111 bp in total) was amplified using the primer pair CP-F/CP-R (Appendix A) and inserted into the pK2-Sur vector to yield a complementary (CP) vector. The constructed CP vector was transformed into Δ*MaPom1*, which was subsequently selected on Czapek–Dox agar supplemented with 20 μg/mL chlorimuron ethyl (Sigma-Aldrich, Bellefonte, PA, USA) and verified by PCR.

### 4.4. Southern Blotting

Genomic DNA (6 µg) extracted from each strain was subjected to enzymatic digestion with NheI and AflII restriction enzymes, and the digested DNA was separated on 1.0% agarose gel and transferred onto a nylon membrane. The specific probe, spanning 523 bp, was derived from the 5′ genomic region of *MaPom1* and amplified using the TF/TR primer pair (Appendix A). Subsequent experiments were conducted according to the instructions of the Roche DIG High Prime DNA Labeling and Detection Starter Kit I (Mannheim Roche, Marnheim, Germany).

### 4.5. Quantitative Reverse Transcription PCR (qRT-PCR)

Total RNA was extracted from conidia using an Ultrapure RNA Kit (with DNase I) (CWBIO, Beijing, China). RNA samples were reverse-transcribed using the PrimeScript RT Reagent Kit with gDNA Eraser (TaKaRa, Beijing, China) according to the manufacturer’s instructions. Gene-specific qRT-PCR primers (Appendix A) were designed using the NCBI website (https://www.ncbi.nlm.nih.gov/ (accessed on 14 February 2022)). qRT-PCR was performed with the SYBR Prime qRT-PCR Set (Bio Ground, Chongqing, China) using a two-step method, as per the manufacturer’s protocol.

### 4.6. Serine/Threonine Kinase Activity Analysis

Total protein was extracted from 0.2 g treated conidia (heat-shock treatment for 0, 3, and 6 h and UV-B treatment for 0, 3, and 4.5 h) using a filamentous fungal protein extraction kit (Bestbio, Shenzhen, China). Protein concentration was measured using a BCA Protein Assay Kit (Beyotime Biotechnology, Nanjing, China) following the manufacturer’s guidelines. The protein concentrations of different samples were adjusted to 10 μg/μL to ensure consistency in enzyme activity measurements, which were conducted using a broad-spectrum serine/threonine kinase assay kit (ELISA, ICP0248, Immune Chem, Vancouver, BC, Canada) [103]. This detection kit is based on the principle of solid-phase enzyme-linked immunosorbent assay (ELISA), using specific peptide substrates of Serine/Threonine kinase (STK) and polyclonal antibodies that can specifically recognize phosphorylated substrates to detect STK activity in the liquid phase. The specific method is as follows: pre-coat the microplate with STK specific peptide substrate, add the test samples to the corresponding well, and start the substrate phosphorylation reaction by adding ATP. After the reaction is complete, a specific antibody is added to the well to bind to a specific phosphorylated substrate, and then the specific antibody is bound to a secondary antibody coupled with peroxidase (HRP). The coupled HRP can act on the substrate of tetramethylbenzidine (TMB) for color development. After the color development process is terminated by an acidic termination solution, the optical density (OD) is measured at 450 nm using Microplate Reader (Berthold Technologies, Winnenden, Germany). The OD_450_ values are directly proportional to the enzyme activity.

### 4.7. Microscopic Investigation and Fluorescence Staining

Conidia were stained with calcofluor white (CFW) under dark conditions [104]. Mature conidia of the WT, Δ*MaPom1*, and CP strains were collected to prepare spore suspensions (2 × 10^6^ conidia/mL), which were then added to 1/4 SDAY medium and incubated for approximately 20 h. DNA damage in WT, Δ*MaPom1*, and CP was observed using a DAPI staining protocol [105]. Conidial suspensions (50 µL aliquots, 1 × 10^7^ conidia/mL) were plated onto 1/4 SDAY plates for UV-B and heat shock treatments. All samples were observed and photographed using a fluorescence microscope (Nikon Eclipse Ci-E, Tokyo, Japan). (The percentage of cells with nuclear damage was calculated as the ratio of nuclear-damaged cells to the total number of cells).

### 4.8. Conidial Germination and Conidial Yield Assays

Conidial germination was performed according to a previously established protocol [106]. Aliquots of 50 µL conidial suspensions of WT, Δ*MaPom1*, and CP (1 × 10^7^ conidia/mL) were plated onto 1/4 SDAY medium for continuous incubation at 28 °C. Conidial germination statistics were recorded every 2 h. Conidial yield assays were performed according to previously established protocols [107]. Aliquots of 2 µL conidial suspensions (1 × 10^6^ conidia/mL) were dipped onto 1/4 SDAY medium for subsequent incubation at 28 °C. Three independent replicates of each individual strain were harvested at three-day intervals from the third day until the fifteenth day. Each experimental procedure was performed three times to ensure consistency and reliability.

### 4.9. Virulence Assays

Bioassays were performed according to procedures outlined in previous studies [108]. Aliquots of 4 μL conidial suspensions (1 × 10^7^ conidia/mL) prepared in paraffin oil were dipped onto the pronotum of the insects following a previously described protocol (control insects were inoculated with 4 μL paraffin oil). Each experimental group had three replicates, each with 30 insects, and the mortality rates of the locusts were recorded at 12 h intervals. The entire experiment was independently replicated three times.

### 4.10. Stress Tolerance Assays

Stress tolerance was measured using existing methods [109]. For the UV-B tolerance assay, aliquots of 50 μL suspensions (1 × 10^7^ conidia/mL) were evenly distributed onto 1/4 SDAY medium and subjected to UV-B irradiation (1350 mW/m^2^) for 1.5, 3, 4.5, and 6 h. The samples were subsequently cultured at 28 °C and harvested at the specified times to record the germination rates. For the thermotolerance assay, 50 μL aliquots of the same suspensions were exposed to 45 °C for 3, 6, 9, and 12 h, spread on 1/4 SDAY medium, and harvested at the indicated times to record the germination rates. To assess sensitivity to abiotic stress, 2 μL aliquots of suspensions (1 × 10^6^ conidia/mL) were placed onto 1/4 SDAY and 1/4 SDAY medium containing various stressors: 6 mM hydrogen peroxide (H_2_O_2_), 1 M sorbitol (SOR), 1 M sodium chloride (NaCl), 500 μg/mL Congo red (CR), 50 mg/mL calcofluor white (CFW), and 0.01% sodium dodecyl sulfate (SDS). The plates were incubated at 28 °C for 6 days [110].

### 4.11. Trehalose Extraction and Quantification Assays

Trehalose extraction was performed according to the protocol provided in the trehalose content assay kit (Solarbio, Beijing, China) [111]. Mature conidia (0.1 g) were extracted using 1 mL extraction solution. The fungal conidia were then crushed by ultrasonic treatment (200 W, 10 s intervals, 3 s of ultrasonication, repeated 30 times) at 4 °C. The mixture was centrifuged at 8000× *g* for 10 min at room temperature to obtain the supernatant. A 0.25 mL aliquot of the supernatant was transferred to an EP tube and mixed with 1 mL working solution. The samples were incubated at 95 °C for 10 min. The absorbance (A) was measured at 620 nm, and the procedure was repeated three times.

### 4.12. Digital Gene Expression Profiling (DGE)

Total RNA was extracted from the conidia of the WT and Δ*MaPom1* strains cultured on 1/4 SDAY plates for 3 days. RNA sequencing was performed using an Illumina HiSeq 2000 platform with three biological replicates (Beijing Genomics Institute, Chongqing, China). Differentially expressed genes (DEGs) were identified based on criteria of with log2(∆*MaPom1*-3 d/WT-3 d) ≥ 2 and a false discovery rate (FDR) ≤ 0.001. Gene Ontology (GO) and Kyoto Encyclopedia of Genes and Genomes (KEGG) pathway enrichment analyses were conducted to classify and annotate the obtained DEGs.

### 4.13. Data Analysis

Data analyses were performed using the SPSS software (version 17.0) (SPSS Inc., Chicago, IL, USA). Graphs were generated with GraphPad Prism version 9.0 for Windows (GraphPad Software, San Diego, CA, USA, www.graphpad.com (accessed on 20 September 2024)). One-way analysis of variance (ANOVA) and Tukey’s test were used to assess the significance of differences between experimental groups. Significance levels were defined as *p* < 0.05 (*), *p* < 0.01 (**), and *p* < 0.001 (***), with “ns” indicating no significant differences.

## 5. Conclusions

In summary, our study revealed a new function of *MaPom1* in the regulation of stress tolerance. Specifically, we identified a potential regulatory mechanism of *MaPom1* in stress tolerance. *MaPom1* regulated the expression of genes related to DNA damage repair, resulting in reduced UV-B tolerance, and decreased thermotolerance was related to the expression of HSP genes. Overall, *MaPom1* contributes to both UV-B tolerance and thermotolerance via distinct pathways. These findings highlight the importance of *MaPom1* in stress tolerance and provide new insights into the regulatory mechanisms that enable fungi to withstand abiotic stresses. This study also lays the groundwork for enhancing the efficacy of entomopathogenic fungi for insect control.

## Figures and Tables

**Figure 1 ijms-25-11860-f001:**
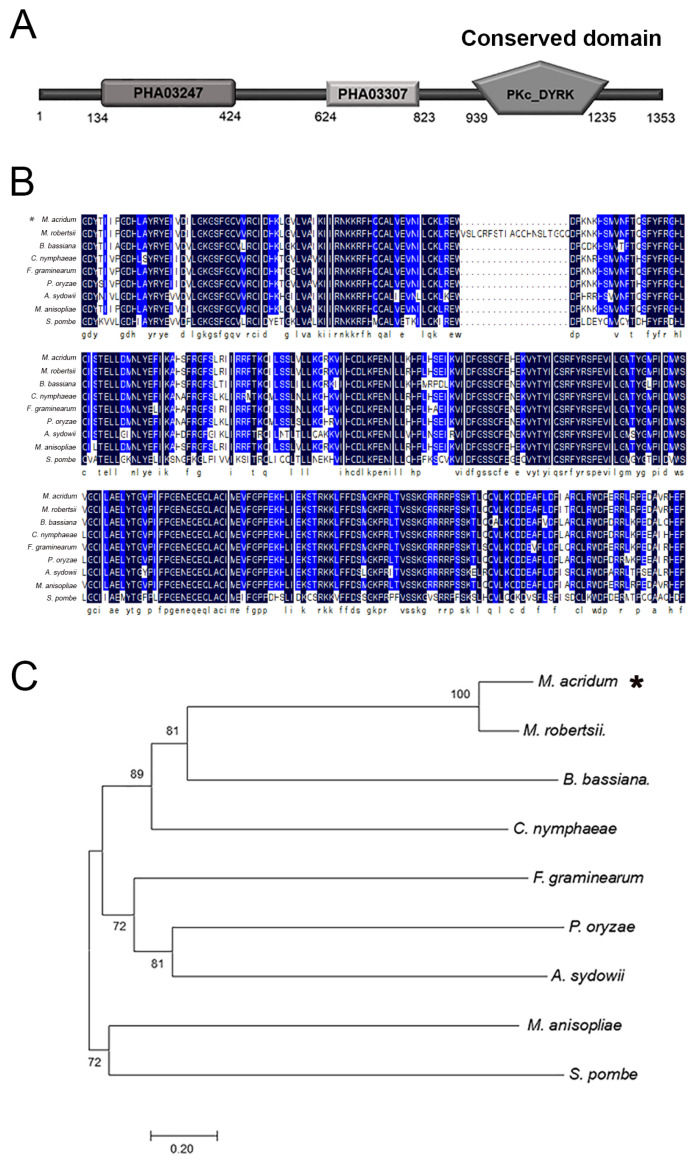
Characteristics of MaPom1. (**A**) Main domains of the Pom1 protein. PKc_DYRK: A serine/threonine protein kinase conserved domain. (**B**) Alignment of the conserved domains of the Pom1 protein with homologous proteins from other fungi, including *Metarhizium acridum* (XP_007808957.1), *Metarhizium robertsii* (EXV02279.1), *Beauveria bassiana* (KAF1729971.1), *Colletotrichum nymphaeae* (KXH60025.1), *Fusarium graminearum* (EYB25727.1), *Pyricularia oryzae* (ELQ36107.1), *Aspergillus sydowii* (OJJ58174.1), *Metarhizium anisopliae* (KAF5127571.1), and *Schizosaccharomyces pombe* (NP_592974.1). The asterisk represents the *M*. *acridum* used in this study. (**C**) Phylogenetic tree was reconstructed using the neighbor-joining method by the MEGA 7.0 software.

**Figure 2 ijms-25-11860-f002:**
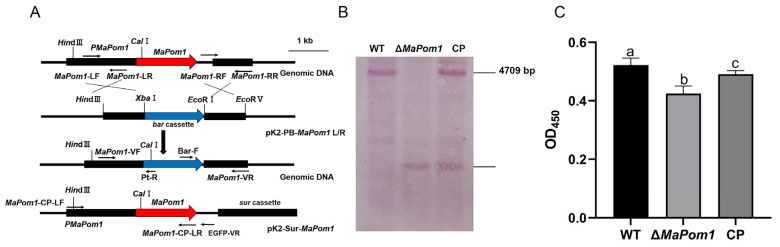
Design and verification of *MaPom1* knockout and complementary strains. (**A**) Construction of homologous recombination vectors and the random insertion principle in *M. acridum*. (**B**) Southern blot verification of the wild-type and mutant strains. Wild-type and mutant strains were digested with NheI and AflII. The probe sequence on the *MaPom1* gene was amplified using TF/TR (Appendix A), yielding a product with a length of 523 bp. WT: wild-type strain; Δ*MaPom1*: knockout strain; CP: complementary strain. (**C**) Determination of serine/threonine protein kinase activity. The OD_450_ values are directly proportional to the enzyme activity. Error bars represent standard deviations based on three independent replicates. The different lowercase letters indicate significant differences by one-way ANOVA and Tukey’s test.

**Figure 3 ijms-25-11860-f003:**
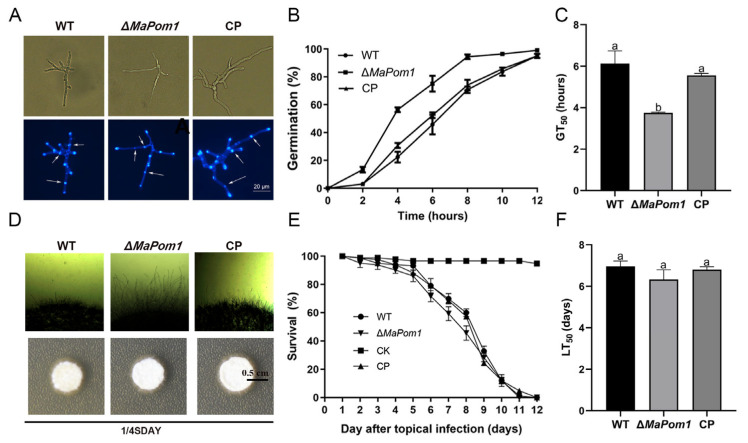
Conidial germination, mycelial growth, and virulence assays. (**A**) Mycelial morphology on 1/4 SDAY medium after 18 h of incubation at 28 °C. Mycelia were stained with CFW (Calcofluor white) to visualize the septa; the arrows represent hyphal septa. (**B**) Conidial germination rates on 1/4 SDAY medium at 28 °C. (**C**) Half-germination time (GT_50_) of each strain. (**D**) Observation of hyphal length and colony morphology on 1/4 SDAY medium for 3 d at 28 °C. (**E**) Survival rates of locust inoculated with 4 µL conidial suspensions. (**F**) Half-lethal time (LT_50_) after topical inoculation with conidia from each strain. Error bars represent standard deviations based on three independent replicates. The different lowercase letters indicate significant differences by one-way ANOVA and Tukey’s test.

**Figure 4 ijms-25-11860-f004:**
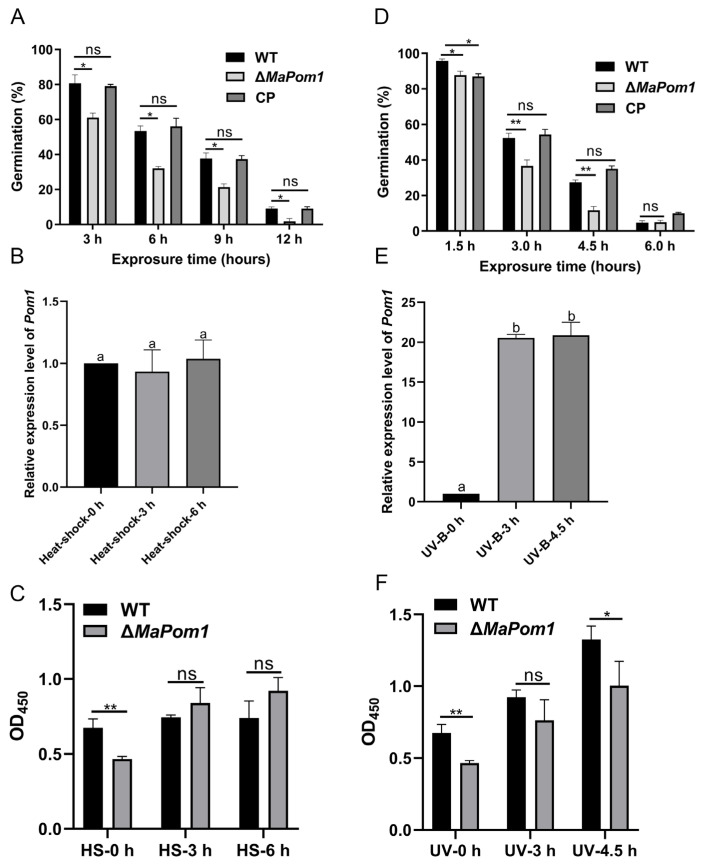
Stress tolerance assays. (**A**) Germination rates on 1/4 SDAY medium after 20 h of cultivation following heat shock treatment for 3, 6, 9, and 12 h. (**B**) Relative expression levels of *MaPom1* after heat shock treatment (45 °C) for 0, 3, and 6 h. (**C**) Serine/threonine protein kinase activity after heat shock treatment for 0, 3, and 6 h. The OD_450_ values are directly proportional to the enzyme activity. (**D**) Germination rates on 1/4 SDAY medium after 20 h of cultivation following UV-B irradiation for 1.5, 3, 4.5, and 6 h. (**E**) Relative expression levels of *MaPom1* after UV-B irradiation (1350 mW/m^2^) for 0, 3, and 4.5 h. (**F**) Serine/threonine protein kinase activity after UV-B irradiation for 0, 3, and 4.5 h. The OD_450_ values are directly proportional to the enzyme activity. Error bars represent standard deviations based on three independent replicates. Asterisks indicate significant differences at *p* < 0.05 (*), *p* < 0.01 (**), while “ns” indicates no significant differences, and the different lowercase letters indicate significant differences by one-way ANOVA and Tukey test.

**Figure 5 ijms-25-11860-f005:**
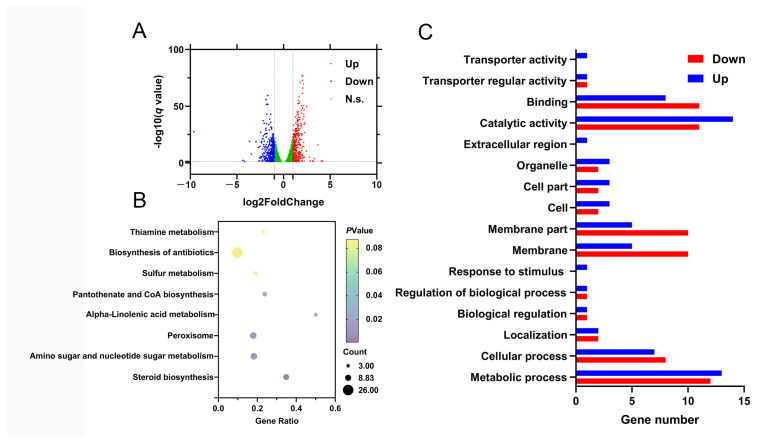
DEGs analysis of WT and Δ*MaPom1*. (**A**) Volcano plot of DEGs from Δ*MaPom1*-3 d vs. WT-3 d. (**B**) GO annotation of DEGs from Δ*MaPom1*-3 d vs. WT-3 d. (**C**) KEGG annotation of DEGs from Δ*MaPom1*-3 d vs. WT-3 d.

**Figure 6 ijms-25-11860-f006:**
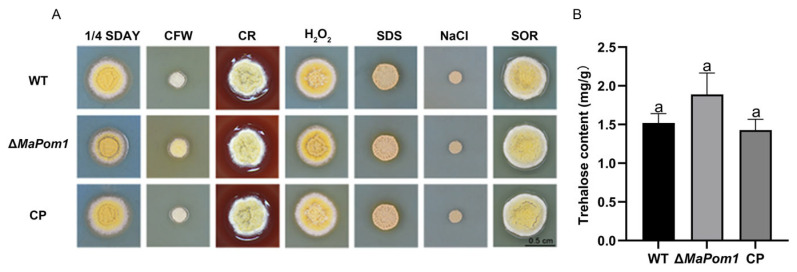
Stress tolerance and trehalose content analysis. (**A**) Fungal colonies on 1/4 SDAY medium supplemented with 0.01% *w/v* SDS, 500 μg/mL CR, 1 mol/L NaCl, 50 μg/mL CFW, 6 mM H_2_O_2_, and 1 mol/L SOR. (**B**) Trehalose content in 15-day-old conidia. Error bars represent standard deviations based on three independent replicates. The same lowercase letters indicate no significant difference.

**Figure 7 ijms-25-11860-f007:**
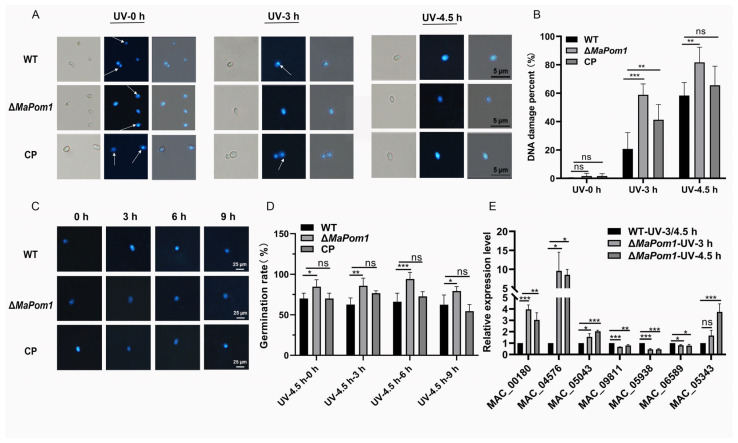
Analysis of DNA damage and repair after UV-B irradiation. (**A**) DNA staining with DAPI after UV-B irradiation for different periods. “UV-0 h” denotes no UV-B treatment, “UV-3 h” denotes UV-B treatment for 3 h, and “UV-4.5 h” denotes UV-B treatment for 4.5 h. Arrows represent stained nuclear DNA. (**B**) Percentage of DNA damage after UV-B treatment. (**C**) Degree of DNA damage repair in conidia cultured on 1/4 SDAY medium for 0, 3, 6, and 9 h at 28 °C after UV-B irradiation for 4.5 h. (**D**) Percentage of DNA damage after 4.5 h of UV-B treatment and subsequent culture on 1/4 SDAY medium for 0, 3, 6, and 9 h at 28 °C. “UV-4.5 h-0 h” indicates conidia without recovery on 1/4 SDAY medium after 4.5 h of UV-B treatment, “UV-4.5 h-3 h” indicates conidia recovery on 1/4 SDAY medium for 3 h after UV-B treatment for 4.5 h, “UV-4.5 h-6 h” indicates conidia recovery on 1/4 SDAY medium for 6 h after UV-B treatment for 4.5 h, and “UV-4.5 h-9 h” indicates conidia recovery on 1/4 SDAY medium for 9 h after UV-B treatment for 4.5 h. (**E**) Relative expression levels of DNA damage repair-related genes after UV-B treatment for 3 and 4.5 h. “WT-UV-3/4.5 h” indicates conidia of WT after UV-B treatment for 3, 4.5 h. “Δ*MaPom1*-UV-3 h” and “Δ*MaPom1*-UV-4.5 h” represent conidia of WT subjected to UV-B treatment for 3 h or 4.5 h, respectively. The relative expression levels of different genes in WT were normalized to 1 after different treatments. All the images were acquired at a fixed exposure time. Error bars represent standard deviations based on three independent replicates. Asterisks indicate significant differences at *p* < 0.05 (*), *p* < 0.01 (**), *p* < 0.01 (***), while “ns” indicates no significant differences by one-way ANOVA and Tukey’s test.

## Data Availability

The authors confirm that the data supporting the findings of this study are available within this article and its Appendix A. The RNA seq data has been uploaded to NCBI, the ac-cession numbers for these SRA data: SRX11118927-SRX11118932.

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
