# Peer review of "MaPom1, a Dual-Specificity Tyrosine Phosphorylation-Regulated Kinase, Positively Regulates Thermal and UV-B Tolerance in *Metarhizium acridum"

_ijms, 2024, doi:10.3390/ijms252211860_

Round 1
Reviewer 1 Report
Comments and Suggestions for Authors
An interesting article on the function of MaPom1 in the entomopathogenic fungus Metarhizium acridum and its regulatory role in stress tolerance. Thermal and UV-B tolerance of conidia were significantly decreased in the MaPom1 disrupting strain, while conidia yield and virulence remained unchanged. RNA-Seq analysis was performed. The effect of heat shock on MaPom1 expression and enzyme activity was investigated and evaluated. The percentage of nuclear damage in ΔMaPom1 under UV-B irradiation was higher than in the wild-type and complemented strains. MaPom1 contributed to thermal and UV-B tolerance by regulating the expression of heat shock proteins and DNA damage repair genes.
The introduction is accessible in a way that simply introduces the research topic. Figures are legible. Statistical tests were performed, but ANOVA values were not recorded. There are no statistically significant differences marked on the graphs (Figure 4). References contain a large review of the literature.
In summary, the study revealed a novel function of MaPom1 in the regulation of stress tolerance, providing new insights into the regulatory mechanisms that enable fungi to withstand abiotic stresses, and may in the future increase the efficacy of entomopathogenic fungi in controlling insects.
Author Response
Comments 1: The introduction is accessible in a way that simply introduces the research topic. Figures are legible. Statistical tests were performed, but ANOVA values were not recorded. There are no statistically significant differences marked on the graphs (Figure 4).
Response 1: Thanks for your comments. We have taken your comments into consideration and have made several changes to the revised manuscript.
We applied the one-way ANOVA analysis and Tukey test in this study. The ANOVA values have been included in the revised manuscript, specifically in lines 123-125, 154-155, 157-158, 162-164, 171-172, 190-191, 193-194, 200-201, 202-204, and 268-269.
Reviewer 2 Report
Comments and Suggestions for Authors
The authors of the manuscript entitled "MaPom1, a dual-specificity tyrosine phosphorylation-regulated kinase, contributes to thermal and UV-B tolerance in Metarhizium acridum" performed very interesting investigations of the role of the MaPom1 kinase in the response of the Metarhizium acridum to the heat and UV stresses. They indicated this enzyme's role in the fungus's reaction to both stresses in different ways. The methods applied are well chosen, and the data support conclusions. Therefore, I recommend publishing this manuscript after the authors correct it according to the minor comments I provide below.
1. In the Methods (lanes 415-418), detailed information on how the kinase activity was assayed needs to be included. The indicated reference – 103, is a conference proceedings material, and I could not find a way to access the full description. Therefore, any researcher willing to repeat the same assay may experience the same problem. That is why the procedure should be briefly described, and the principle of the assay should be explained, even if the kit is commercial. Moreover, suppose this kit is an ELISA kit to detect kinases. In that case, the results do not show the enzyme activity but the relative level of kinase protein in the assayed sample. That is a very important difference, as a small number of enzyme molecules can have, in particular conditions, higher activity than a large number of the same protein in other conditions.
2. Lanes 122-123. How do you explain that the knockout mutant has any enzyme activity? Can other kinases replace the MaPom1?
3. Here, I believe the authors wrongly described the plots in Figure 4. Specifically:
Lane 176 -177. The indicated figure shows the germination percentage, not the MaPom1 mRNA expression levels.
Lane 178 -180. I see no differences in bars representing 0, 3, and 6h of heat shock in Figure 4C. Moreover, according to the description of the Y-axis, this plot indicates the impact of heat shock on the relative expression of Pom1 mRNA.
Lane 181. In Figure 4D, the bars indicate the Pom1 mRNA expression levels, which I assume as a fold change in ΔMaPom1 compared to WT using the proper reference gene.
Lane 183. The description fits Figure 4D, not 4E.
Also, descriptions under Figure 4 do not fit, except for 4A and 4D:
4B - The time course fits Fig. 4C
4C - The time course fits Fig. 4E
4D - The time course fits Fig. 4B
4E - The time course fits Fig. 4D
4. The discussion is generally nicely written, but I still need to include at the end the description of the model of how Pom1 acts under heat and UV stresses.
Author Response
Comments 1: In the Methods (lanes 415-418), detailed information on how the kinase activity was assayed needs to be included. The indicated reference-103, is a conference proceedings material, and I could not find a way to access the full description. Therefore, any researcher willing to repeat the same assay may experience the same problem. That is why the procedure should be briefly described, and the principle of the assay should be explained, even if the kit is commercial. Moreover, suppose this kit is an ELISA kit to detect kinases. In that case, the results do not show the enzyme activity but the relative level of kinase protein in the assayed sample. That is a very important difference, as a small number of enzyme molecules can have, in particular conditions, higher activity than a large number of the same protein in other conditions.
Response 1: We greatly appreciate your comments. We replaced the reference-103 with the original literature of the method. According to your suggestion, we also described the enzyme activity assay method in detail at the “4.6. Serine/Threonine kinase activity analysis” section in the revised manuscript (line 449-468).
Additionally, we used the same protein concentration of these samples under the same conditions in the enzyme activity assay. In revised manuscript, we added the specific concentration in the method section (line 454). Thus, Fig. 2C represents the relative difference in the protein kinase activity among different samples under the same conditions.
Comments 2: Lanes 122-123. How do you explain that the knockout mutant has any enzyme activity? Can other kinases replace the MaPom1?
Response 2: Thanks for your comments. The measured kinase activity in the results represents the total kinase activity in the samples, thus kinase activity in the knockout mutant could be originated from other Serine/Threonine kinase (s).
Yes, there are many other kinases in the strain. On the enzyme activity, the other kinases may replace the MaPom1, but whether the other kinases can replace MaPom1 on the biological function (s) has not studied in our study.
Comments 3: Here, I believe the authors wrongly described the plots in Figure 4. Specifically:
Response 3: We greatly appreciate your comments. There are indeed errors in the labeling in Figure 4. We have corrected the mistakes in figure 4.
Lane 176-177. The indicated figure shows the germination percentage, not the MaPom1 mRNA expression levels.
We have replaced the germination percentage with mRNA expression level of MaPom1.
Lane 178 -180. I see no differences in bars representing 0, 3, and 6h of heat shock in Figure 4C. Moreover, according to the description of the Y-axis, this plot indicates the impact of heat shock on the relative expression of Pom1 mRNA.
Yes, the mRNA relative expression of Pom1 has no differences in Figure 4B (revised manuscript) under heat shock treatments.
Lane 181. In Figure 4D, the bars indicate the Pom1 mRNA expression levels, which I assume as a fold change in ΔMaPom1 compared to WT using the proper reference gene.
The bars indicate the Pom1 mRNA expression levels under different treatments in WT, indicating MaPom1 plays an important role under UV-B treatment in Metarhizium acridum.
Lane 183. The description fits Figure 4D, not 4E.
Also, descriptions under Figure 4 do not fit, except for 4A and 4D:
4B - The time course fits Fig. 4C
4C - The time course fits Fig. 4E
4D - The time course fits Fig. 4B
4E - The time course fits Fig. 4D
Thanks a lot, we have corrected it.
Comments 4: The discussion is generally nicely written, but I still need to include at the end the description of the model of how Pom1 acts under heat and UV stresses.
Response 4: Thanks for your comments. We believed that your suggestions were helpful, so we have added the descriptions of the model for how MaPom1 acts under UV-B (line 383-386) and heat shock (line 402-404) stresses at the end of discussion, respectively.
Reviewer 3 Report
Comments and Suggestions for Authors
The article deeply analyzes MaPom1 (a Pom1 homologous gene) in the entomopathogenic fungus Metarhizium acridum. The practical significance of the study is related to the development of new methods for combating insect species harmful to agriculture. The authors used modern methods of genetic and biochemical research. Most of the conclusions in the article are substantiated. At the same time, the manuscript contains a number of shortcomings.
1. The title of the article should be formulated differently: "Effect of MaPom1 activity on heat and UVB tolerance in Metarhizium acridum" or something similar. The title should describe the effect of something on something, not the result of this effect.
2. The Material and Methods section should be moved before the Results section.
3. Lines 469 and 461 are the same: this is an error.
4. Figure 1C: the data clustering method should be indicated in the figure title.
5. Figure 2C: are the differences between the first and third columns reliable? Here you need to apply the Tukey test above the columns, write letters. In the title of the figure, you need to indicate the repetition of samples. If the repetition is greater than 4, then the results in Figure 1C should be presented as a box analysis (median, first and third quartiles, minimum and maximum values).
6. Figure 3B, 3D: statistical processing by the Tukey test should be carried out for each time point. Above the graphs, you need to mark with different letters the points that are significantly different from each other according to the results of the Tukey test. In the title of the figure, you need to indicate the repetition of samples.
7. Figure 4: If the repetition is greater than 4, then the results in Figure 1C should be presented as a box analysis (median, first and third quartiles, minimum and maximum values). Boxes should be signed with different letters if they are significantly different from each other according to the results of the Tukey test. In the title of the figure, you need to indicate the repetition of samples. On the ordinate axis, you need to write the name of the characteristic with a capital letter, a comma, a unit of measurement. 8. The same applies to Figures 6B, 7B, 7D, 7E. Figure 7 is too small to see anything: what's the point of Figures 7A and 7C if readers can't see anything in them?
9. All fonts in all figures should be approximately equal to the height of the letters in the text of the article. Do not use bold.
10. Figure 5A should be enlarged.
11. References are not allowed in the Results section. Results should not contain elements of Materials and Methods and comparisons of your results with other people's data. These fragments of text should be moved to the Discussion.
12. The Discussion section needs to be rewritten: the authors conducted a large study, presented in several figures and tables. The Discussion should also be divided into subsections, the titles of which should roughly correspond to the groups of figures in the results (4-6 subsections).
Author Response
Comments 1: The title of the article should be formulated differently: "Effect of MaPom1 activity on heat and UV-B tolerance in Metarhizium acridum" or something similar. The title should describe the effect of something on something, not the result of this effect.
Response 1: Thanks for your suggestion. We have revised the title to “MaPom1, a dual-specificity tyrosine phosphorylation-regulated kinase, positively regulates thermal and UV-B tolerance in Metarhizium acridum”.
Comments 2: The Material and Methods section should be moved before the Results section.
Response 2: Thanks for your comments. However, this journal requires the Material and Methods section before the Results section.
Comments 3: Lines 469 and 461 are the same: this is an error.
Response 3: Thanks, we have corrected this error in line 518 and line 526.
Comments 4: Figure 1C: the data clustering method should be indicated in the figure title.
Response 4: Thanks for your comments. We have added the data clustering method in the figure legend at line 134-136.
Comments 5: Figure 2C: are the differences between the first and third columns reliable? Here you need to apply the Tukey test above the columns, write letters. In the title of the figure, you need to indicate the repetition of samples. If the repetition is greater than 4, then the results in Figure 1C should be presented as a box analysis (median, first and third quartiles, minimum and maximum values).
Response 5: We sincerely appreciate the valuable comments. We have taken your comments into consideration and have made several changes in the revised manuscript. Specifically, the differences between the first and third columns are reliable and we have added the ANOVA values to the manuscript to better understand the differences between the first and third columns (line 123-125).
We have applied one-way ANOVA analysis and Tukey test to clarify the differences in Figure 2C, with the differences expressed via lowercase letters above the columns. Additionally, we have added figure legends to provide further information in the revised manuscript (line 145-148).
We sincerely appreciate your input and hope that these changes address your concerns.
Comments 6: Figure 3B, 3D: statistical processing by the Tukey test should be carried out for each time point. Above the graphs, you need to mark with different letters the points that are significantly different from each other according to the results of the Tukey test. In the title of the figure, you need to indicate the repetition of samples.
Response 6: Thanks for your comments. We used the Figure 3B and 3D to depict the trends in germination and host survival over time. To support these findings, we conducted a statistical analysis using one-way ANOVA and Tukey test on half-germination time (GT50) and half-lethal time (LT50) in Figure 3C and 3F. The ANOVA values have been included in the revised manuscript (line 162-164, line 171-172). Additionally, we have added the repetition of samples to the figure legends, specifically in lines 146, 182, 217-218, 276, and 327-328. We hope that these changes address your concerns and improve the clarity of our findings.
Comments 7: Figure 4: If the repetition is greater than 4, then the results in Figure should be presented as a box analysis (median, first and third quartiles, minimum and maximum values). Boxes should be signed with different letters if they are significantly different from each other according to the results of the Tukey test. In the title of the figure, you need to indicate the repetition of samples. On the ordinate axis, you need to write the name of the characteristic with a capital letter, a comma, a unit of measurement. The same applies to Figures 6B, 7B, 7D, 7E.
Response 7: Thanks for your comments. We have used lowercase letters above the columns to represent the differences in Figures 4B, 4E, and 6B, and have added ANOVA values for clarity in the manuscript (line 193-195, line 202-204, line 268-269).
However, we have continued to use asterisks to represent the differences in Figure 4A, 4B, 4E, 4F, 7B, 7D, and 7E, since we only conducted one-way ANOVA at the same time conditions without considering the differences between different times.
Additionally, we have added the repetition of samples to the figure legends and included units of measurement in the figures for clarity.
Comments 8: Figure 7 is too small to see anything: what's the point of Figures 7A and 7C if readers can't see anything in them?
Response 8: Thanks for your comments. The point represents the degree of damage to nuclear DNA, and dispersion represents DNA damage. Additionally, we have enlarged the Figure 7A in the revised manuscript.
Comments 9: All fonts in all figures should be approximately equal to the height of the letters in the text of the article. Do not use bold.
Response 9: Thanks for your comments. We have corrected it.
Comments 10: Figure 5A should be enlarged.
Response 10: Thanks for your comments. Figure 5A represented the trend of differentially expressed genes (different colors), which has been clearly demonstrated. Thus, no need to enlarge it.
Comments 11: References are not allowed in the Results section. Results should not contain elements of Materials and Methods and comparisons of your results with other people's data. These fragments of text should be moved to the Discussion.
Response 11: Thanks for your comments. We have made some modifications and removed the related references from the Results section to the Discussion section. However, we still retained some references cited in the Results section, because the references can provide clues for experiments, explaining why we make this experiment and helping readers to understand the research ideas.
Additionally, we understand that brief descriptions of certain materials or methods in the Results section may be helpful for readers to understand the experimental design and results. As a result, we have included these descriptions as appropriate.
Comments 12: The Discussion section needs to be rewritten: the authors conducted a large study, presented in several figures and tables. The Discussion should also be divided into subsections, the titles of which should roughly correspond to the groups of figures in the results (4-6 subsections).
Response 12: Thanks for your comments. We have re-written the Discussion section in the revised manuscript based on your suggestions. The discussion section has been organized into subsections that correspond to the related results in line 331-404. We hope that these changes will improve the clarity and organization of the revised manuscript, and make it easier for readers to understand our findings.
Round 2
Reviewer 3 Report
Comments and Suggestions for Authors
The article has been sufficiently improved by the authors.